# A Systematic Review of the Reliability and Validity of the Patient Activation Measure Tool

**DOI:** 10.3390/healthcare12111079

**Published:** 2024-05-24

**Authors:** Qin Xiang Ng, Matthias Yi Quan Liau, Yong Yi Tan, Ansel Shao Pin Tang, Clarence Ong, Julian Thumboo, Chien Earn Lee

**Affiliations:** 1Health Services Research Unit, Singapore General Hospital, Singapore 169608, Singapore; 2Saw Swee Hock School of Public Health, National University of Singapore and National University Health System, Singapore 117549, Singapore; 3Lee Kong Chian School of Medicine, Nanyang Technological University, Singapore 308232, Singapore; 4NUS Yong Loo Lin School of Medicine, National University of Singapore, Singapore 117597, Singapore; 5SingHealth Duke-NUS Medicine Academic Clinical Programme, Duke-NUS Medical School, Singapore 169857, Singapore; 6SingHealth Office of Regional Health, Singapore 911532, Singapore

**Keywords:** patient activation, patient activation measure, review, validity

## Abstract

Patient activation, broadly defined as the ability of individuals to manage their health and navigate the healthcare system effectively, is crucial for achieving positive health outcomes. The Patient Activation Measure (PAM), a popularly used tool, was developed to assess this vital component of health care. This review is the first to systematically examine the validity of the PAM, as well as study its reliability, factor structure, and validity across various populations. Following the PRISMA and COSMIN guidelines, a search was conducted in MEDLINE, EMBASE, and Cochrane Library, from inception to 1 October 2023, using combinations of keywords related to patient activation and the PAM. The inclusion criteria were original quantitative or mixed methods studies focusing on PAM-13 or its translated versions and containing data on psychometric properties. Out of 3007 abstracts retrieved, 39 studies were included in the final review. The PAM has been extensively studied across diverse populations and geographical regions, including the United States, Europe, Asia, and Australia. Most studies looked at populations with chronic conditions. Only two studies applied the PAM to community-dwelling individuals and found support for its use. Studies predominantly showed a high internal consistency (Cronbach’s alpha > 0.80) for the PAM. Most studies supported a unidimensional construct of patient activation, although cultural differences influenced the factor structure in some cases. Construct validity was established through correlations with health behaviors and outcomes. Despite its strengths, there is a need for further research, particularly in exploring content validity and differential item functioning. Expanding the PAM’s application to more diverse demographic groups and community-dwelling individuals could enhance our understanding of patient activation and its impact on health outcomes.

## 1. Introduction

The World Health Organization (WHO) defines self-care as “the ability of individuals, families, and communities to promote health, prevent disease, maintain health, and cope with illness and disability with or without the support of a healthcare provider” [1]. At the core of this concept is patient activation, a behavioral notion encompassing an individual’s knowledge, skills, and confidence in managing their health and health care [2]. Patient activation is thought to be integral to self-care and achieving positive population health outcomes, as it empowers patients to play an active role in their health management. Recent research, including a 2022 meta-analysis encompassing nine observational studies, supports the positive impact of high patient activation scores, linking them to decreased visits to emergency departments, fewer hospital admissions, and reduced overall healthcare utilization [3]. 

However, it is estimated that between 11% and 47% of the population possess low levels of activation, making them less likely to adopt healthy behaviors [4,5]. To enhance patient activation, interventions have been developed and show that a positive change in activation levels correlates with improved self-care behavior. The critical step in this process is measuring patient activation, and this is where tools like the Patient Activation Measure (PAM) come into play [6]. 

The PAM, crafted by Hibbard and colleagues, outlines four stages of patient activation [6,7]. These stages range from patients acknowledging their pivotal role in their health care to gaining knowledge and confidence for active participation, translating this confidence and knowledge into action, and maintaining an active role in healthcare despite potential challenges [7]. The PAM, a 13-item survey, assesses an individual’s knowledge, skill, and confidence for self-management. It categorizes patients into four levels based on their activation score, which ranges from a lack of understanding of the need for active participation in health (level one) to challenges in maintaining positive health behaviors over time (level four) [6]. The PAM tool has been validated in various patients with chronic diseases but is less frequently validated in the general population [8,9,10]. 

Given the potential for the widespread application of the PAM in population health systems, it would be important to examine the reliability, validity, strengths, and limitations of the tool. To date, there have not been comprehensive systematic reviews focusing on the PAM tool. Therefore, this review aims to systematically examine the existing literature concerning the validation studies of the PAM tool, including their findings, and the diverse populations and settings in which the PAM has been employed. This would also be carried out in accordance with the latest COSMIN (COnsensus-based Standards for the selection of health Measurement INstruments) guidelines [11] to improve the completeness of the reporting of these studies.

## 2. Methods

The COSMIN [11] and latest PRISMA (Preferred Reporting Items for Systematic reviews and Meta-Analyses) guidelines [12] were referenced for this systematic review. The protocol was registered on PROSPERO (registration number CRD42024529845).

A systematic literature search was conducted to identify published studies that have used the PAM questionnaire using combinations of the keywords: [patient activation OR PAM] AND [health care OR patient care]. The search was performed in MEDLINE, EMBASE, and Cochrane Library from database inception up till 1 October 2023. The full search strategy for the various databases can be found in the Appendix A. The title and abstract of the retrieved searches were screened by two independent researchers (from M.Y.Q.L., Y.Y.T., or Q.X.N.), and the duplicates were removed.

Full texts were retrieved for the articles that met the inclusion criteria: (1) original studies with quantitative or mixed-method methodology, (2) specifically focusing on the PAM-13 or PAM-10 (or translated versions) and reporting the psychometric properties of the tool, and (3) published in English. Commentaries, case reports, case series, and review articles were excluded as well. All disagreements during the screening process were resolved via discussion with the senior author (Q.X.N.). Information was extracted from the selected studies, including author details, year of publication, study design, population characteristics, methodologies used for validation, main findings, and limitations to meet the objectives of this review. The outcomes of the included studies were then descriptively synthesized to arrive at a broader understanding of the validation and psychometric properties of the PAM across different populations.

The included studies were graded according to the COSMIN checklist [13,14] by M.Y.Q.L., A.S.P.T. and Q.X.N. The COSMIN checklist evaluates the methodology quality of each study in nine aspects: (1) content validity, (2) structural validity, (3) internal consistency, (4) cross-cultural validity/measurement invariance, (5) reliability, (6) hypotheses testing for construct validity, and (7) responsiveness. A four-point scale is used to assess each domain (poor, fair, good, or excellent). An overall score for methodological quality in each domain was determined by taking the lowest grade provided for any item within the category. 

Given that this is a review of the existing literature and no primary data collection or human participants were directly involved, prior ethical approval was not required.

## 3. Results

### 3.1. Literature Retrieval

A total of 3007 abstracts were retrieved from the systematic search. After removal of duplicates, 1886 studies remained. After title and abstract screening, 363 articles remained and were sought for full-text review. Six full texts could not be retrieved despite best efforts and manual library search. Of the remaining 357 studies, 304 were not validation or reliability studies pertaining to the PAM, 5 used variations of the PAM e.g., caregiver-PAM, 5 were abstracts or conference proceedings, and 4 did not have English translation. A total of 39 manuscripts [6,8,9,15,16,17,18,19,20,21,22,23,24,25,26,27,28,29,30,31,32,33,34,35,36,37,38,39,40,41,42,43,44,45,46,47,48,49,50] were finally included (Figure 1). 

Table 1 summarizes the key characteristics of the studies and their principal findings regarding reliability, validity, and factor analysis, arranged by their geographic regions according to the WHO regional classification [51].

### 3.2. Populations Sampled

The Patient Activation Measure (PAM) has been extensively studied across diverse populations and geographical regions, including the United States, Europe, Asia, and Australia. In terms of the distribution, there were 13 studies from the United States [6,15,20,22,26,27,32,33,40,44,46]; 3 studies from Germany [18,19,49] and the Netherlands [25,39,43]; 2 studies from Canada [29,41], Israel [34,35], Malaysia [17,24], Norway [37,38], and Singapore [8,9]; and 1 each from Australia [21], China [48], Denmark [36], Hungary [50], India [28], Iran [47], Italy [23], Korea [16], Portugal [31], and Turkey [30]. These studies primarily focused on adults, including specific groups like those with chronic conditions, cancer, heart disease, and mental health issues. The PAM was effective in measuring activation in populations with chronic conditions, with studies showing its ability to differentiate based on health behaviors and management. Notably, only two studies applied the PAM to the general populace and found support for its use among community-dwelling individuals [8,50].

### 3.3. Validity and Reliability Studies

The reliability of the PAM, predominantly assessed using Cronbach’s alpha, demonstrates generally high internal consistency across different studies and populations. The values of Cronbach’s alpha ranged mostly above 0.80, indicating strong reliability. A few studies, such as those by Fowles et al. [22] and Zeng et al. [48], report particularly high reliability scores (α > 0.90). This consistency is crucial, as it suggests the PAM tool consistently measures patient activation regardless of the sample population.

Most studies report a single-factor structure for the PAM, suggesting that it measures a unidimensional construct. However, a few studies like those by Lin et al. [33] in the United States and Zakeri et al. [47] in Iran identified multifactor structures, indicating potential cultural or contextual variations in how patient activation is conceptualized or manifested.

The construct validity of the PAM is supported through various methods, including principal component analysis (PCA) and correlations with other measures. The studies often show moderate-to-strong correlations between PAM scores and relevant health behaviors or outcomes. For instance, Eyles et al. [21] found correlations with depressive symptoms and health-related quality of life. Similarly, studies by Bomba et al. [18] and Stepleman et al. [46] found significant correlations between PAM scores and measures of internal and external locus of control and multiple sclerosis self-efficacy, respectively.

While most studies accepted the original content validation by Hibbards et al. [6,26] as sufficient content validation for implementation in their own cohorts, where locally validated, it appears to be robust. For example, Kosar et al. in Turkey highlighted high content validity indices, suggesting that the items of the PAM are well received and considered relevant by the participants [30].

### 3.4. Cross-Cultural Adaptations and Translations

In other countries in which English may not be the working language, the PAM was translated to a language that was spoken or written by the majority. Across the included studies that translated the PAM and subsequently validated the translated version, all studies followed the prescribed methodology for the cross-cultural adaptation of questionnaires, namely forward–backward translation, qualitative assessment by an expert panel, and pilot testing on a small sample representative of the target population [52]. Translated versions of the PAM, such as in Turkey, Portugal, and Iran, indicate that the tool is adaptable in different linguistic and cultural contexts, although this sometimes affects its factor structure.

### 3.5. Identified Issues

While the PAM is generally reliable across diverse populations, certain subgroups might not be as accurately represented. For instance, Hibbard et al. [26] noted lower reliability for specific subgroups, such as those with no chronic illness, older adults, and those with lower income and education. As highlighted in Table 2, in our appraisal of methodological quality, many studies did not assess content validity in accordance with the COSMIN guidelines. While the reliability and construct validity are well established, the lack of a comprehensive content validity assessment in some studies could question the tool’s comprehensiveness in covering all aspects of patient activation, especially since cultural context does impact on people’s interpretation of questionnaire items. Most studies focused on correlations between PAM scores and other health-related measures to establish construct validity. However, this approach may not fully explore all facets of the patient activation construct, such as psychological, social, and behavioral components. Moreover, some studies indicated the presence of DIF, suggesting that certain items may not have an equivalent meaning across different groups or contexts.

## 4. Discussion

Based on the findings of this systematic review, the PAM is a reliable and valid tool for measuring patient activation across various populations. Its adaptability to different languages and contexts, along with its strong correlation with key health outcomes, underscores its utility in both clinical and research settings. The tool’s ability to capture the essence of a patient’s engagement in their health care makes it a valuable asset in patient-centered care models as patient activation is increasingly acknowledged as a key determinant of health outcomes, and it also lends itself to action-oriented insights.

This review systematically synthesizes findings from various studies to evaluate the validity and applicability of the PAM across different populations and settings. High internal consistency, as evidenced by Cronbach’s alpha values predominantly over 0.80, reinforces the tool’s reliability. This uniformity underscores the PAM’s robustness in measuring patient activation across various demographics and conditions. This reliability is vital for the tool’s use in both clinical settings and large-scale population health studies.

Construct validity, established through correlations with other health measures, demonstrates the PAM’s effectiveness in capturing the essence of patient engagement. This aspect is crucial for tailoring interventions and policies to enhance patient activation and, consequently, health outcomes. Studies like Bomba et al. [18] and Eyles et al. [21] illustrate how PAM scores correlate with meaningful health-related outcomes such as depressive symptoms, quality of life, and locus of control. These correlations are crucial for establishing the relevance of the PAM in predicting and influencing health behaviors and outcomes.

While content validity is less frequently reported, where available, the studies that do address this aspect, like Kosar et al. [30], show that the PAM’s items are considered relevant and reflective of the patient activation construct. Cross-cultural adaptation of the PAM was successfully implemented and validated across different countries, with good representation of European and Asian languages, demonstrating its flexibility and applicability globally. The adherence and complete reporting of the translation methodology also ensured the validity of the translated instrument, thus avoiding problems with conceptual, item, and semantic equivalence [53]. Furthermore, the high levels of reliability, as measured with Cronbach’s alpha, across these translated versions of the PAM seem to imply a uniform understanding of a patient’s activity and disease self-management. However, the emergence of multifactor structures in some cultural contexts indicates that patient activation may be conceptualized differently across cultures, although whether the organization of items into the factors concurred with theoretically acceptable domains was not discussed in the respective reports [17,24,47,48]. This finding underscores the need for cultural sensitivity and adaptability in the application of the PAM because cultural context does impact on people’s interpretation of questionnaire items, and hence future studies may be directed toward understanding the cross-cultural understanding of patient activation and its impact on PAM outcomes.

Beyond establishing reliability and construct validity, validation studies of the PAM have delved into its clinical utility. The instrument’s ability to predict health-related behaviors, outcomes, and healthcare utilization is a critical aspect of its practical application. The research consistently indicates that higher levels of patient activation, as measured by the PAM, are associated with more proactive health management behaviors and improved health outcomes [54]. The predictive validity of the PAM is exemplified by its capacity to anticipate patient behaviors, such as proactive health management and adherence to treatment plans [55,56]. Moreover, higher PAM scores have consistently been associated with improved health outcomes, including reduced healthcare utilization and improved overall quality of life [57,58,59]. This predictive power enables healthcare providers to identify individuals at risk for suboptimal outcomes and tailor interventions to meet specific patient needs. Additionally, the clinical utility of the PAM extends to resource allocation and personalized intervention strategies, making it an indispensable asset for fostering patient engagement, shared decision-making, and ultimately improving the efficiency and effectiveness of healthcare delivery [10,60,61]. This broad applicability supports the instrument’s ability to measure patient activation as a universal construct, possibly transcending specific health conditions and patient demographics.

Patient activation can be conceptualized as an element under the broader frameworks of patient self-management, which also encompasses concrete actions taken toward bettering the health of oneself and interactions with the care team. Wagner’s chronic care model organizes related concepts to describe the stakeholders and factors that participate in chronic disease care, specifically that of an informed, activated patient with productive interactions with a proactive care team against the context of a supportive community [62]. This indicates the ongoing paradigmatic shift away from a paternalistic care model toward a patient–physician partnership in care [63,64,65,66]. The latter then relies on active communication and engagement from the attending physician, so as to activate the patients toward self-management of disease and uptake of health behaviors [67,68]. Thus, measuring patient activation through tools such as the PAM is paramount in the care models of today.

Another tool that was recently developed and validated, which sought to measure patient activation, is the Consumer Health Activation Index (CHAI) [69]. This tool was developed as the authors positioned the PAM as being less appropriate for administration in populations with lower health literacy based on item readability. Moreover, the proprietary nature of the PAM restricts its use. The CHAI is a 10-item questionnaire on a 6-point Likert scale, with a recommended single-factor solution for simpler interpretation and parsimony in a clinical setting. The CHAI has been validated in other populations, such as in Australia and South Korea, and used in various settings [12,70,71,72]. No validation study comparing the reliability, reading level, or differential of both instruments head-on has been conducted and is thus much warranted. This will inform the indication of use for the respective instruments and ensure reciprocal validity between the two scales.

In sum, several notable limitations in the PAM studies were identified. Firstly, there is a notable scarcity of PAM validation studies involving the general population [8,50], and the tool’s variable reliability among specific subgroups—such as older adults and individuals from lower socio-economic backgrounds—casts doubt on its universal applicability. As countries increasingly focus on social prescribing and more effectively addressing the social needs of their populations [73,74], exploring and understanding this aspect becomes a critical area for research. Second, the lack of comprehensive reporting on content validity in some studies poses a challenge in fully understanding the scope of the PAM tool. As aforementioned, the variability in factor structures across different translations suggests a need for further research to understand these differences better. Future studies should aim to address the identified limitations, including the exploration of content validity more comprehensively (as outlined by the COSMIN guidelines) and examining the tool’s applicability in broader and more diverse populations as health care moves toward activating not just patients but individuals in the community.

## 5. Conclusions

The PAM is a reliable and valid tool for assessing patient engagement in health care. A total of 39 validation and reliability studies were reviewed, and the studies demonstrate the application of the PAM across various cultural contexts, as well as the PAM’s ability to correlate with health outcomes relevant to patient-centered health care. Expanding its validation among the general population and other settings would be crucial steps in fully realizing its potential in improving population-wide healthcare delivery and outcomes.

## Figures and Tables

**Figure 1 healthcare-12-01079-f001:**
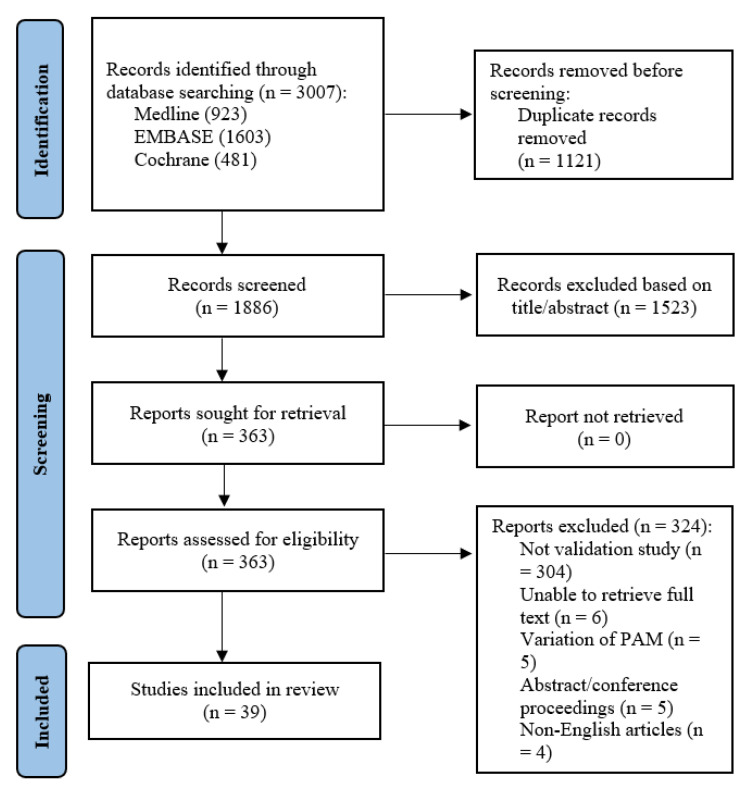
PRISMA flowchart showing the literature search process.

**Table 1 healthcare-12-01079-t001:** Key characteristics of studies reviewed and their findings regarding reliability, validity, and factor analysis (arranged geographically).

Author, Year	Country	Population	Sample Size (N)	PAM Version	Factor Structure	Reliability	Content Validity	Construct Validity
**Region of the Americas (*n* = 15)**								
Acquati, 2022 [15]	United States	Adults ≥25 years, diagnosed with cancer	504	PAM-10	NR	Cronbach’s α = 0.81	NA	NR
Christiansen, 2020 [20]	United States	Adults in fall prevention program	343	PAM-13	NR	Cronbach’s α = 0.87 (pre and post)	CVI (NR)	NR
Fowles, 2009 [22]	United States	People on health promotion programs	625	PAM-13	1 factor	Cronbach’s α = 0.90	NA	Related to healthy behaviors, readiness-to-change
Hibbard, 2004 [6]	United States	Convenience samples of patients and a national probability sample	1515	PAM-13	NR	High level of reliability across subgroups	Expert panel consensus	Associated with better health, lower healthcare usage
Hibbard, 2005 [26]	United States	Original PAM-22 validation study participants	1515	PAM-13	1 factor	Lower reliability for some subgroups	NA	Strong link with activation scores, preventive behaviors
Hung, 2013 [27]	United States	Patients from rural areas	812	PAM-13	1 factor	PSI = 2.36, Cronbach’s α = 0.85	NA	Small correlations with CAHPS, moderate with SM subscales
Lightfoot, 2021 [32]	United States	Patients with CKD, not treated with dialysis	942	PAM-13	1 factor	Cronbach’s α = 0.925	NA	DIF observed for specific items and CKD stages
Lin, 2023 [33]	United States	Family caregivers of patients with chronic illness	277	PAM-10	2 factors	Cronbach’s α = 0.86	NA	Higher activation associated with less depression, better adherence to care
O’Malley, 2018 [40]	United States	Survivors of early-stage breast and prostate cancer	325	PAM-13 (modified)	NR	Cronbach’s α = 0.89	NA	NR
Prey, 2016 [42]	United States	Hospitalized cardiology and oncology patients	100	PAM-13	NR	Cronbach’s α = 0.81	NA	Correlation with PROMIS Global Health
Schmaderer, 2015 [44]	United States	Hospitalized patients with multimorbidity	313	PAM-13	1 factor	Cronbach’s α = 0.88	CVI = 0.91	Correlation with PACIC, PROMIS-29 depression, physical functioning
Skolasky, 2011 [45]	United States	Older adults with multimorbidity	855	PAM-13	NR	Cronbach’s α = 0.87	NA	NR
Stepleman, 2010 [46]	United States	Patients with multiple sclerosis	199	PAM-13	NR	Cronbach’s α = 0.88	NA	Correlation with MSSE, BDI-II, MSQOL
Kephart, 2019 [29]	Canada	People with neurological conditions	738	PAM-13	NR	Cronbach’s α = 0.884	NA	Modest convergent validity with other measures
Packer, 2015 [41]	Canada	Patients with neurological conditions	722	PAM-13	1 factor	Cronbach’s α = 0.87	NA	Correlation with HUI-3, MCS, PCS, SLIQ
**Western Pacific Region (*n* = 3)**								
Ahn, 2015 [16]	Korea	Patients with OA from community health posts	270	PAM-13 (Korean)	1 factor	Cronbach’s α = 0.88, interitem correlation = 0.34	Expert panel consensus	PCA revealed one dimension, 57.5% total variance
Eyles, 2020 [21]	Australia	Patients on an OA management program	217	PAM-13	1 factor	Cronbach’s α = 0.92	NA	Correlation with depressive symptoms, health-related quality of life
Zeng, 2019 [48]	China	Patients with hypertension and/or T2DM	509	PAM-13 (Chinese)	4 factors	Cronbach’s α = 0.920	NA	NR
**South-East Asian Region (*n* = 5)**								
Bahrom, 2020 [17]	Malaysia	Metabolic syndrome in primary care	130	PAM-13 (Malay)	2 factors	Overall Cronbach α = 0.79	CVI = 1.00	NR
Hashim, 2020 [24]	Malaysia	Patients with T2DM	130	PAM-13 (Malay)	3 factors	Overall Cronbach α = 0.87	Expert panel consensus	NR
Ge, 2022 [8]	Singapore	Respondents to the Population Health Index study	824	PAM-13	1 factor	Cronbach’s α = 0.82	NA	High health confidence correlated with higher PAM scores
Ngooi, 2017 [9]	Singapore	Patients with IHD or heart failure	270	PAM-13	1 factor	Cronbach’s α = 0.86	NA	Correlation with PHQ-9, SSE
Kapoor, 2020 [28]	India	Patients with DM	200	PAM-13 (Kannada)	NR	Cronbach’s α = 0.84	NA	NR
**European Region (*n* = 13)**								
Bomba, 2018 [18]	Germany	Adolescents with chronic conditions	586	PAM-13 (German)	1 factor	Cronbach’s α = 0.79, rtt = 0.68	NA	Positive correlation with internal LOC, negative with external LOC
Brenk-Franz, 2013 [19]	Germany	German-speaking primary care patients	508	PAM-13 (German)	1 factor	Cronbach’s α = 0.84	NA	NR
Zill, 2013 [49]	Germany	Members of health funds with chronic disease	4018	PAM-13 (German)	1 factor	Cronbach’s α = 0.88	NA	NR
Graffigna, 2015 [23]	Italy	Patients with chronic disease	519	PAM-13 (Italian)	1 factor	Cronbach’s α = 0.88	NA	Moderate-to-strong inter-rest correlations
Kosar, 2019 [30]	Turkey	Internal medicine polyclinic patients	130	PAM-13 (Turkish)	1 factor	Cronbach’s α = 0.81	High content validity index	Strong construct validity, confirmed by factor analyses
Laranjo, 2018 [31]	Portugal	Outpatients with T2DM	193	PAM-13 (Portuguese)	1 factor	Item reliability = 0.97	Expert panel consensus and CVI = 0.98	DIF analysis comparing English and Portuguese versions
Melby, 2021 [37]	Norway	Patients with substance use disorders or schizophrenia spectrum disorders	57	PAM-13 (Norwegian)	NR	SUDs: Cronbach’s α = 0.75–0.84; Schizophrenia spectrum: α = 0.87–0.81	NA	NR
Moljord, 2015 [38]	Norway	Patients with mental illness	273	PAM-13 (Norwegian)	2 factors	Overall Cronbach α = 0.87	NA	NR
Maindal, 2009 [36]	Denmark	Participants in a primary care health education with dysglycemia	358	PAM-13 (Danish)	1 factor	Cronbach’s α = 0.89	NA	No significant DIF in subgroups
Nijman, 2014 [39]	Netherlands	Dutch Healthcare Consumer Panel	1432	PAM 13-(Dutch)	NR	Cronbach’s α = 0.83	NA	NR
Rademakers, 2012 [43]	Netherlands	Chronic illness or disability	1837	PAM 13-(Dutch)	NR	Cronbach’s α = 0.88	NA	Correlation with SBSQ-D
Hendrikx, 2018 [25]	Netherlands	Population health program participants	3120	PAM-13 (Dutch)	NR	Cronbach α = 0.89–0.90	NA	Comparable results with MCS, K10, PAM13
Zrubka, 2022 [50]	Hungary	General population >40 years	779	PAM-13 (Hungarian)	1 factor	Cronbach’s α = 0.77	Expert assessment during translation process	Correlation with eHEALS
**Eastern Mediterranean Region (*n* = 3)**								
Magnezi, 2014 [34]	Israel	Nationally representative population	203	PAM-13 (Hebrew)	NR	Cronbach’s α = 0.77	NA	Correlation with self-efficacy, SF-12, PHQ-9
Magnezi, 2014 [35]	Israel	Primary care patients	278	PAM-13 (Hebrew)	NR	Cronbach’s α = 0.78	NA	Correlation with PHQ-9, SF-12
Zakeri, 2023 [47]	Iran	Chronically ill patients	438	PAM-13 (Persian)	3 factors	Cronbach’s α = 0.88	S-CVI = 0.91; I-CVI = 0.63–1.00	Correlation with PIH, SWLS

Abbreviations: BDI, Beck depression inventory; CKD, chronic kidney disease; CVI, content validity index; DIF, differential item functioning; eHEALS, electronic health literacy scale; IHD, ischemic heart disease; LOC, locus of control; MSQOL, Leeds multiple sclerosis quality of life; NA, not assessed; NR, not reported; OA, osteoarthritis; PAM, Patient Activation Measure; PCA, principal component analysis; PIH, partner in health measure; PROMIS, patient reported outcomes measurement information system; PSI, person separation index; SSE, Stanford self-efficacy for managing chronic disease six-item scale; SUD, substance use disorder; SWLS, satisfaction with life scale; T2DM, type 2 diabetes mellitus.

**Table 2 healthcare-12-01079-t002:** Methodological quality of each study per measurement property.

Study	Content Validity	Structural Validity *	Internal Consistency	Cross-Cultural Validity/Measurement Invariance	Reliability	Hypotheses Testing for Construct Validity	Responsiveness
Acquati, 2022 [15]	Poor	Fair	Excellent	-	Poor	-	-
Ahn, 2015 [16]	Good	Excellent	Excellent	-	Good	-	-
Bahrom, 2020 [17]	Excellent	Good	Excellent	-	Good	-	-
Bomba, 2018 [18]	Fair	Excellent	Excellent	Fair	Good	Excellent	Excellent
Brenk-Franz, 2013 [19]	Fair	Good	Excellent	-	Good	Excellent	Excellent
Christiansen, 2020 [20]	Fair		Excellent	Poor	Good	-	Good
Eyles, 2020 [21]	Poor	Excellent	Excellent	-	Good	Good	Good
Fowles, 2009 [22]	Poor	Excellent	Excellent	-	Good	Poor	Poor
Ge, 2022 [8]	Poor	Excellent	Excellent	-	Good	Excellent	Excellent
Graffigna, 2015 [23]	Fair	Excellent	Excellent	-	Fair	-	-
Hashim, 2020 [24]	Fair	Good	Excellent	-	Fair	-	-
Hendrikx, 2018 [25]	Poor	Excellent	Excellent	-	Poor	Excellent	Excellent
Hibbard, 2004 [6]	Fair	Excellent	Excellent	-	Good	Good	Good
Hibbard, 2005 [26]	Poor	Excellent	Poor	-	Poor	Excellent	Excellent
Hung, 2013 [27]	Poor	Excellent	Poor	-	Fair	Excellent	Excellent
Kapoor, 2020 [28]	Fair	Poor	Excellent	-	Fair	-	-
Kephart, 2019 [29]	Poor	Fair	Excellent	-	Poor	Excellent	Excellent
Kosar, 2019 [30]	Fair	Fair	Excellent	-	Good	-	-
Laranjo, 2018 [31]	Excellent	Good	Poor	-	Excellent	-	-
Lightfoot, 2021 [32]	Poor	Excellent	Excellent	Fair	Poor	-	Good
Lin, 2023 [33]	Poor	Excellent	Excellent	-	Good	Excellent	Excellent
Magnezi, 2014 [34]	Fair	Excellent	Excellent	-	Fair	Excellent	Excellent
Magnezi, 2014 [35]	Poor	Poor	Excellent	-	Poor	Excellent	Excellent
Maindal, 2009 [36]	Fair	Excellent	Excellent	Fair	Poor	-	
Melby, 2021 [37]	Poor	Poor	Excellent	Fair	Excellent	-	
Moljord, 2015 [38]	Poor	Good	Excellent	-	Excellent		Excellent
Ngooi, 2017 [9]	Poor	Fair	Excellent	-	Good	Excellent	Excellent
Nijman, 2014 [39]	Poor	Poor	Excellent	-	Poor	Fair	Fair
O’Malley, 2018 [40]	Poor	Poor	Good	Fair	Poor	-	Fair
Packer, 2015 [41]	Poor	Excellent	Excellent	-	Fair	Excellent	Excellent
Prey, 2016 [42]	Poor	Poor	Excellent	-	Good	Excellent	Excellent
Rademakers, 2012 [43]	Fair	Poor	Excellent	Fair	Fair	Excellent	Excellent
Schmaderer, 2015 [44]	Excellent	Fair	Excellent	-	Fair	Excellent	Excellent
Skolasky, 2011 [45]	Poor	Poor	Excellent	-	Good	Good	Good
Stepleman, 2010 [46]	Poor	Good	Excellent	-	Good	Excellent	Excellent
Zakeri, 2023 [47]	Fair	Excellent	Excellent	-	Excellent	Excellent	Excellent
Zeng, 2019 [48]	Poor	Fair	Excellent	-	Good	-	-
Zill, 2013 [49]	Poor	Excellent	Excellent	-	Fair	-	-
Zrubka, 2022 [50]	Fair	Excellent	Excellent	-	Good	Excellent	Excellent

* Structural validity refers to the degree to which the scores of the scale used are an adequate indication for dimensionality of the construct, attribute, or factor being measured.

## Data Availability

The data that support the findings of this study are available from the corresponding author upon reasonable request.

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
