# Peer review of "A Systematic Review of the Reliability and Validity of the Patient Activation Measure Tool"

_healthcare, 2024, doi:10.3390/healthcare12111079_

Round 1
Reviewer 1 Report
Comments and Suggestions for Authors
very interesting research but it can be improved:
it is desirable to regist it in PROSPERO
provide a quality evaluation of included articles (like rob2)
Author Response
Thank you for the comments and suggestions.
- We have now added the PROSPERO registration ID in the methods section, "The protocol was registered on PROSPERO (registration number CRD42024529845)."
- The methodological quality of each study per measurement property was appraised using the COSMIN guidelines, "Included studies were graded according to the COSMIN checklist [13]. It evaluates the methodology quality of each study in nine aspects: (1) Content Validity, (2) Structural Validity, (3) Internal Consistency, (4) Cross-cultural validity/measurement invariance, (5) Reliability, (6) Hypotheses testing for construct validity and (7) Responsiveness. A four-point scale is used to assess each domain (poor, fair, good or excellent). An overall score for methodological quality in each domain was determined by taking the lowest grade provided for any item within the category." The findings are reported in Table 2 (in the results section).
Reviewer 2 Report
Comments and Suggestions for Authors
The manuscript describes a systematic review of the Patient Activation Measure (PAM), a tool designed to assess patient engagement in healthcare. A total of 39 validation and reliability studies were filtered and reviewed, across diverse cultural contexts and its correlation with relevant health outcomes. The review underscores the PAM's reliability, validity, and utility in both clinical and research settings. The manuscript emphasizes ongoing research and application of the PAM. It also highlights the need for further validation among the general population and in various healthcare settings to fully leverage its potential for improving population healthcare delivery and outcomes. To provide a more comprehensive and informative analysis, the authors could include the following analysis:
Any potential limitations, biases, or gaps in these reviewed studies?
Author Response
Thank you for the comments. We have now added in the discussion section that, "several notable limitations in the PAM studies were identified. Firstly, there is a notable scarcity of PAM validation studies involving the general population [8,49], and the tool’s variable reliability among specific subgroups—such as older adults and individuals from lower socio-economic backgrounds—casts doubt on its universal applicability. As countries increasingly focus on social prescribing and more effectively addressing the social needs of their populations [72,73], exploring and understanding this aspect becomes a critical area for research. Second, the lack of comprehensive reporting on content validity in some studies poses a challenge in fully understanding the scope of the PAM tool. As aforementioned, the variability in factor structures across different translations suggests a need for further research to understand these differences better. Future studies should aim to address the identified limitations, including the exploration of content validity more comprehensively (as outlined by the COSMIN guidelines) and examining the tool’s applicability in broader and more diverse populations as health care moves towards activating not just patients but individuals in the community."
Reviewer 3 Report
Comments and Suggestions for Authors
REVIEW REPORT FOR THE STUDY “A SYSTEMATIC REVIEW OF THE VALIDITY OF THE PATIENT ACTIVATION MEASURE TOOL”
Journal: Healthcare
The paper "A Systematic Review of the Validity of the Patient Activation Measure Tool", performs an analysis to better identify the validity of the PAM (Patient Activation Measure), across various populations.
Title and summary. The title and abstract express well the object of study, objectives, and results of the article.
Structure of the article. The contents are well organized and they adhere to the IMRaD structure. It includes a theoretical framework of the research problem and, focusing on the opportunity of the study, it must be said that it is useful work since it covers one of the problems resulting from a health care system.
Materials and methods.
Regarding the material and methods section, the methodology is tailored to the object of study and the objectives and is explained in a transparent manner while it has been validly applied to guarantee the results.
Nevertheless, it would be interesting to indicate which internal validity and possible bias assessment guide has been followed, for example, the standard one published by the Cochrane Collaboration. This phase must be carried out by at least two independent researchers (the authors indicate that this has been done) but it must also be carried out in a masked form to avoid evaluation biases (and this is not indicated by the authors; it would be convenient for the readers to indicate this).
In that sense, it is recommended to evaluate the quality of evidence following the GRADE method [Schünemann H, Brożek J, Guyatt G, Oxman A. GRADE Handbook. Handbook for grading the quality of evidence and the strength of recommendations using the GRADE approach. 2013: https://gdt.gradepro.org/app/ handbook/handbook.html].
Also, it would be interesting to indicate which guide was used for the analysis of the research designs. for example, the STROBE checklist. [Von Elm E, Altman DG, Egger M, Pocock SJ, Gøtzsche PC, Vandenbroucke JP. STROBE Initiative. The Strengthening the Reporting of Observational Studies in Epidemiology (STROBE) statement: guidelines for reporting observational studies. J Clin Epidemiol. 2008, Apr;61(4):344-9. PMID: 18313558]
I also suggest indicating the descriptive checklist for articles included in a scoping study on which the process of analyzing the selected articles was based (example, the one proposed by Malo M and Robert E. [Malo M, Robert E. Proposition: canevas pour une Scoping Study.2011. Available: http://equiperenard.ca/fr/grille-danalys?e.html]
It should be indicated whether the data synthesis has been performed according to which procedure for meta-aggregation of data (e.g., that of the JBI guidelines for systematic reviews of qualitative studies). [Lockwood C, Porrit K, Munn Z, Rittenmeyer L, Salmond S, Bjerrum M et al Chapter 2: Systematic reviews of qualitative evidence. In: Aromataris E, Munn Z, editors. Joanna Briggs Institute Reviewer’s Manual. The Joanna Briggs Institute; 2017.].
Results.
The results are significant and they are presented in an adequate and understandable way not only through narration but also with self-explained tables and figures that are also well elaborated in terms of presentation. The results justify and relate to the objectives and methods and the results are of sufficient interest.
Discussion.
The discussion appropriately compares the study results with other works, highlighting the main study findings.
Overall, this is an interesting study and should be considered for publication in Healthcare, once the main proposed revisions have been resolved.
Author Response
Thank you for the comments and suggestions.
- The findings are reported in accordance with COSMIN (COnsensus-based Standards for the selection of health Measurement INstruments) guidelines and also the PRISMA guidelines. A copy of the completed PRISMA checklist was appended to the supplementary material.
- In terms of the appraisal of study quality, the methodological quality of each study per measurement property was appraised using the COSMIN guidelines, "Included studies were graded according to the COSMIN checklist [13]. It evaluates the methodology quality of each study in nine aspects: (1) Content Validity, (2) Structural Validity, (3) Internal Consistency, (4) Cross-cultural validity/measurement invariance, (5) Reliability, (6) Hypotheses testing for construct validity and (7) Responsiveness. A four-point scale is used to assess each domain (poor, fair, good or excellent). An overall score for methodological quality in each domain was determined by taking the lowest grade provided for any item within the category." The findings are reported in Table 2 (in the results section).
Reviewer 4 Report
Comments and Suggestions for Authors
Study that analyzes the reliability of the Patient Activation Measure Tool by means of a systematic review study. The manuscript presented is very ambitious in its objective, something that the authors were subsequently unable to achieve.
The Introduction section should make a solid case for the aim of the study. This should address the gaps in knowledge about the existence of these tools or, alternatively, the existing knowledge gaps with the Patient Activation Measure Tool. Please justify the need for this study. Why is there a need to know more about this?.
The manuscript mentions throughout the validation of the PAM in different populations and diseases. However, the table shows in most studies that content validity was not reported. This requires the authors to stop and rewrite the objective of the study. Similarly, the title would also need to be modified.
The registration number required for systematic reviews is missing from the manuscript. Please include it in the manuscript (methods section).
The fact of including 39 studies as selected studies makes it difficult for the reader to understand the results. I suggest that the authors restructure the table, grouping the studies for example by health-disease conditions or by geographic region. Also, suggest being able to eliminate those studies that do not provide sufficient information in terms of the validity of the questionnaire.
It would be of interest for the authors to include analyses on the methodological quality of the studies included as well as their degree of recommendation with some of the tools most commonly used for this purpose.
Author Response
Thank you for the helpful suggestions and comments.
- While the PAM has been validated in numerous studies, its application has predominantly been within specific demographic and patient groups. There is a general paucity of research concerning its validation and reliability in the general population, particularly across varied cultural and socio-economic contexts. This raises questions about the universal applicability of PAM, as different populations may exhibit unique challenges and behaviors in health management that the tool might not currently capture. We have now emphasized in the introduction section that, "Given the potential for the widespread application of the PAM in population health systems, it would be important to examine the reliability, validity, strengths and limitations of the tool. To date, there has not been comprehensive systematic reviews focusing on the PAM tool. Therefore, this review aims to systematically examine the existing literature concerning the validation studies of the PAM tool, including their findings, and the diverse populations and settings where the PAM has been employed."
- We thank the reviewer for the opportunity to improve upon this manuscript. In the original study conducted by Hibbard et al. (2004; PMID: 15230939), face/content validity was assessed using expert consensus and patient focus groups. However, the numerical content validity index (CVI) was not explicitly reported. Instead, experts and patients reviewed the domains and subdomains to establish the relevance and comprehensiveness of the PAM tool. In some studies, such as Acquati et al. (2021; PMID: 34510365), the rigorous development through expert consensus and patient interviews, as described in the original study, was taken as a sufficient content validation for implementation. Therefore, studies do not mention a content validation method. We agree with the reviewer that this is a potential limitation to the study and that caution should be taken when interpreting the results. To improve the accuracy of this manuscript, we have since made changes to Table 1 and modified the results (pg. 7 line. 167-169; pg. 8, line 186-190) and discussion (pg. 10, line. 280-287) and highlighted where content validity was either not reported (NR) or not assessed (NA) at all. In addition, we agree with the reviewer with respect to improving upon the listed objectives (pg. 2, line. 66-68) and title (pg. 1, line. 2-3) of this study to reflect more accurately what our study is communicating the readership and hence, have retitled our manuscript accordingly.
- We have now added the PROSPERO registration ID in the methods section, "The protocol was registered on PROSPERO (registration number CRD42024529845)."
- We thank the reviewer for the opportunity to ensure readability of this study. We fully agree with the reviewer that a geographical re-arrangement of the current table 1 would aid in making the manuscript easier to understand. This also enables the reader to interpret the findings of studies implementing PAM in countries with close geographical proximity. We have made the aforementioned changes to Table 1 in our current revised manuscript. In addition, we have also taken the reviewer’s advice to review the appropriateness of the included studies in relation to our study objectives. We have decided to retain all studies but instead report where content validity was lacking as this is an important step as highlighted in the COSMIN guidelines.
- The methodological quality of each study per measurement property was appraised using the COSMIN guidelines, "Included studies were graded according to the COSMIN checklist [13]. It evaluates the methodology quality of each study in nine aspects: (1) Content Validity, (2) Structural Validity, (3) Internal Consistency, (4) Cross-cultural validity/measurement invariance, (5) Reliability, (6) Hypotheses testing for construct validity and (7) Responsiveness. A four-point scale is used to assess each domain (poor, fair, good or excellent). An overall score for methodological quality in each domain was determined by taking the lowest grade provided for any item within the category." The findings are now reported in Table 2 (in the results section).
Round 2
Reviewer 1 Report
Comments and Suggestions for Authors
paper is now better, I suggest include exclusioin/inclusion criteria
Author Response
1. We thank the reviewer for the kind comments provided. We agree with the reviewers’ comments and have made the appropriate changes to our methodology section (pg. 2 line. 88-92) to improve the manuscript, "Full texts were retrieved for articles that met the inclusion criteria: (1) original studies with quantitative or mixed methods methodology, (2) specifically focusing on the PAM-13 (or translated versions) and reporting the psychometric properties of the tool, and (3) published in English. Commentaries, case reports, case series and review articles were excluded as well."
Reviewer 4 Report
Comments and Suggestions for Authors
We thank the authors for addressing the recommendations made in the first review. They should indicate whether they have followed any previously established guidelines for the division according to geographical region, or whether this was their own elaboration.
Finally, in table 2, they should answer the item ‘structural validity’ of the study by Christiansen, 2020 [19].
Author Response
- We are grateful for the opportunity to improve upon our manuscript in the last revision. The geographical regions were divided according to the WHO region classification (https://www.who.int/countries/). We have. To accurately convey this, we made the appropriate changes to our results section (pg. 3 line. 121-123) to improve the manuscript.
- Structural validity is defined as the degree to which scores of a scale are an adequate indication for dimensionality of the construct, attribute or factor being measured (PMID: 20494804). We have amended this definition into table 2 (pg. 10 line. 202-203) to clarify this point of confusion. We thank the reviewer for the opportunity to improve upon the clarity of this manuscript. We hope that the changes made have sufficiently address these areas of concerns prior to publication of this study.